# META-LEARNING FOR VARIATIONAL INFERENCE

## ABSTRACT

Variational inference (VI) plays an essential role in approximate Bayesian inference due to its computational efficiency and general applicability. Crucial to the performance of VI is the selection of the divergence measure in the optimization objective, as it affects the properties of the approximate posterior significantly. In this paper, we propose a meta-learning algorithm to learn (i) the divergence measure suited for the task of interest to automate the design of the VI method; and (ii) initialization of the variational parameters, which reduces the number of VI optimization steps drastically. We demonstrate the learned divergence outperforms the hand-designed divergence on Gaussian mixture distribution approximation, Bayesian neural network regression, and partial variational autoencoder based recommender systems.

## 1 INTRODUCTION

Bayesian inference provides a powerful tool for probabilistic modelling of data, however, exact Bayesian inference is often intractable. Therefore practical Bayesian inference often resort to approximations, and various approximate inference methods have been developed. Variational Inference (VI) (Jordan et al., 1999; Zhang et al., 2018) approximates the intractable target distribution through optimization of a tractable distribution. Compared to Markov chain Monte Carlo (MCMC), VI is biased but more computationally efficient, making it particularly suitable to large-scale models in deep learning such as Bayesian neural networks and deep generative models.

VI approximates the target distribution by minimizing a divergence objective. Different divergence metrics essentially define a different inference algorithm which leads to different properties of the approximation. Therefore the selection of this divergence is one of the crucial factors of making VI successful. The most widely used divergence measure is $\text{KL}(q||p)$ where $p$ is the target distribution and $q$ is the approximate distribution. However, using this KL divergence for VI has been criticized for under-estimating the uncertainty (Bishop, 2006; Blei et al., 2017; Wang et al., 2018a), which leads to poor model performance when uncertainty estimation is essential. Many alternative divergence measures have been proposed for VI to alleviate this issue (Minka et al., 2005; Hernández-Lobato et al., 2016; Li & Turner, 2016; Csiszár et al., 2004; Bamler et al., 2017; Wang et al., 2018a), which provide better bias and variance trade-offs and lead to better predictive results with more accurate uncertainty estimation.

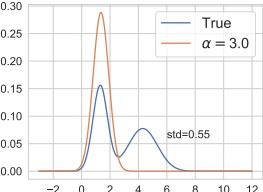 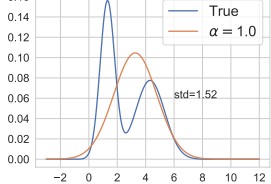 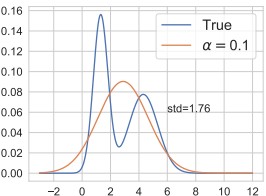

Figure 1: An illustration of approximating distributions on a Gaussian mixture by different value of $\alpha$ (defined in Eq.(3)). "std" is the standard deviation of the Gaussian approximation.

However, as illustrated by Figure 1, this line of work has also shown that the optimal divergence can vary depending on tasks and an undesired divergence objective can lead to mediocre performance

(Li & Turner, 2016; Depeweg et al., 2016). Unfortunately, choosing a suitable divergence objective for a specific task is challenging as it requires a thorough understanding of (i) the shape of the target distribution; (ii) the desirable properties of the approximate distribution (e.g. mass-covering or mode-seeking); and (iii) the bias-variance trade-off between the tightness and the variance of the Monte Carlo estimate of the variational bound. A crucial question remains to be addressed in order to make VI a success for a wide range of applications: can we automatically choose a suitable divergence which are tailored to specific type of tasks?

To answer this question, we propose meta-learning for variational inference (*meta-VI*) which utilizes the advantages of meta-learning to improve approximate Bayesian inference. Meta-learning is to design a learner based on several training tasks that can generalize well to future tasks (Naik & Mammone, 1992; Thrun & Pratt, 2012; Hochreiter et al., 2001). Our meta-VI learns an inference algorithm that is tailored to the problem of interest. Additionally, meta-VI can provide a good initialization of the variational parameters which reduces the training time remarkably. We summarize our contributions as the following:

- We developed a general framework for meta-learning variational inference (Section 3.1), which can choose divergence objective automatically by meta-learning. Specifically, we derive meta-VI first for $\alpha$-divergence based VI, then extend it to $f$-divergence family and provided a novel parameterization of $f$-divergence which is then used in meta-VI.

- In addition to meta-learning the divergence objective, we further combine meta-VI with meta-learning parameters for the variational distribution (Section 3.2). This method provides initialization for fast adaptation and automatically learns the divergence to optimize.

- We demonstrate the efficacy of our approach on various experimental settings (Section 4). On both the Gaussian mixture distribution approximation and Bayesian neural network regression tasks, meta-VI significantly out-performs all baseline methods. On a large-scale, real-world example, meta-VI also improves performance when applied to partial variational auto-encoder based recommender systems.

## 2 Preliminaries

Considering a dataset $\mathcal{D} = \{x_n\}_{n=1}^N$ and a $\theta$-parameterized model. Bayesian inference requires computing the posterior over $\theta$ given the dataset $\mathcal{D}$: $p(\theta|\mathcal{D}) = \frac{p(\mathcal{D}|\theta)p(\theta)}{p(\mathcal{D})}$. The exact posterior $p(\theta|\mathcal{D})$ is generally intractable, therefore one needs to resort to approximation with a tractable approximate posterior $q(\theta) \approx p(\theta|\mathcal{D})$. Typically the approximation posterior $q(\theta)$ is obtained by minimizing a divergence, e.g. Variational Inference (VI) applications often minimize $\text{KL}(q(\theta)||p(\theta|\mathcal{D}))$. This turns Bayesian inference into an optimization task (divergence minimization). In practice, due to the intractability of $p(\mathcal{D})$, VI alternatively maximizes an equivalent objective called the *variational lower bound*:

$$\mathcal{L}_{\text{VI}} = \mathbf{E}_q \left[ \log \frac{p(\mathcal{D}, \theta)}{q(\theta)} \right] = \log p(\mathcal{D}) - \text{KL}(q||p) \tag{1}$$

**Renyi's $\alpha$-divergence** $\alpha$-divergence is a rich family that includes many common divergence as special cases (Minka, 2001; Hernández-Lobato et al., 2016; Li & Turner, 2016). In this paper, we focus on Renyi's definition (Rényi et al., 1961; Li & Turner, 2016) rather than the others (Amari, 2012; Tsallis, 1988) due to its unified expression of gradient for all finite $\alpha$. The benefit of having a unified expression will be seen in Section 3.1. Renyi's $\alpha$-divergence is defined by

$$D_\alpha(p||q) = \frac{1}{\alpha - 1} \log \int p(\theta)^\alpha q(\theta)^{1-\alpha} d\theta, \quad \alpha > 0, \alpha \neq 1, \tag{2}$$

where $D_\alpha(p||q) \to \text{KL}(p||q)$ when $\alpha \to 1$. Similar to maximizing the variational lower bound when the divergence objective is $\text{KL}(q||p)$, one can maximize the *variational Renyi bound* (VR bound) derived from Renyi's $\alpha$-divergence:

$$L_\alpha(q; \mathcal{D}) = \frac{1}{1 - \alpha} \log \mathbf{E}_q \left[ \left( \frac{p(\theta, \mathcal{D})}{q(\theta)} \right)^{1-\alpha} \right] = \log p(\mathcal{D}) - D_\alpha(q||p) \tag{3}$$

The reparameterization trick (Salimans et al., 2013; Kingma & Welling, 2013) is commonly used in practice for gradient ascent based optimization of the VR bound Eq.(3), where sampling $\theta \sim q_\phi(\theta)$ is conducted by first sampling $\epsilon \sim p(\epsilon)$ from a simple distribution independent with the variational parameter $\phi$ (e.g. Gaussian) then parameterizing $\theta = h_\phi(\epsilon)$. Using the reparameterization trick, the VR bound Eq.(3) becomes $L_\alpha(q_\phi; \mathcal{D}) = \frac{1}{1-\alpha} \log \mathbf{E}_\epsilon \left[ \left( \frac{p(h_\phi(\epsilon), \mathcal{D})}{q_\phi(h_\phi(\epsilon))} \right)^{1-\alpha} \right]$. The expectation is usually computed by Monte Carlo (MC) approximation. The gradient of VR bound w.r.t. $\phi$ after MC approximation with $K$ particles is

$$\nabla_\phi L_\alpha(q_\phi; x) = \sum_{k=1}^{K} \left[ w_{\alpha,k} \nabla_\phi \log \frac{p(h_\phi(\epsilon_k), x)}{q(h_\phi(\epsilon_k))} \right] \tag{4}$$

where $w_{\alpha,k} = \left( \frac{p(h_\phi(\epsilon_k), x)}{q(h_\phi(\epsilon_k))} \right)^{1-\alpha} \Big/ \sum_{k=1}^{K} \left[ \left( \frac{p(h_\phi(\epsilon_k), x)}{q(h_\phi(\epsilon_k))} \right)^{1-\alpha} \right]$. When $\alpha = 0$ the weights $w_{\alpha,k} = 1/K$ and the corresponding gradient (4) becomes an unbiased estimate of the gradient of the variational lower bound (1).

As shown in Figure 1, approximate inference with different $\alpha$-divergences results in different variational distributions. Minka et al. (2005) and Li & Turner (2016) also showed the optimal $\alpha$-divergence may vary for different tasks and datasets, and in practice it is difficult to choose an optimal $\alpha$-divergence *a priori*.

**$f$-divergence** $f$-divergence defines a more general family of divergences (Csiszár et al., 2004; Minka et al., 2005). It can be defined using a twice differentiable convex function $f : \mathbb{R}_+ \to \mathbb{R}$ (Csiszár et al., 2004):

$$D_f(p||q_\phi) = \mathbf{E}_{\theta \sim q_\phi} \left[ f \left( \frac{p(\theta)}{q_\phi(\theta)} \right) - f(1) \right]. \tag{5}$$

This family includes KL-divergences in both directions which can be seen by taking $f(t) = -\log t$ for $\mathrm{KL}(q||p)$ and $f(t) = t \log t$ for $\mathrm{KL}(p||q)$. It also contains $\alpha$-divergences which takes $f(t) = \frac{t^\alpha}{\alpha(\alpha-1)}$ for $\alpha \in \mathbb{R} \setminus \{0, 1\}$. Although $f$-divergence family is very rich due to the usage of arbitrary twice differentiable convex function, it requires significant expertise to design a suitable $f$ function for a specific task.

# 3 META-VI

The goal of meta-learning a variational inference algorithm is to learn a divergence objective, so that the resulting VI algorithm produces an approximate distribution with desired properties on a certain type of tasks. To achieve this, we first construct a learnable divergence family, then design a meta-loss function that gives guidance for updating the divergence.

Assume we have $M$ training tasks $T_1, ..., T_M$ sampled from an underlying task distribution $p(\mathcal{T})$. Each task has its own probabilistic model $p_{T_i}(\theta_i, \mathcal{D}_{T_i})$. Let $D_\eta(\cdot||\cdot)$ be the learnable divergence parameterized by $\eta$, then for each task the approximate posterior $q_{\phi_i}(\theta_i)$ is computed by minimizing $D_\eta(p_{T_i}(\theta_i|\mathcal{D}_{T_i})||q_{\phi_i}(\theta_i))$. In the rest of the paper we also write $D_\eta(q_{\phi_i}, T_i) = D_\eta(p_{T_i}(\theta_i|\mathcal{D}_{T_i})||q_{\phi_i}(\theta_i))$ for brevity. During meta-training, we define a meta-loss function $\mathcal{J}(q_{\phi_i}, T_i)$ which is optimized w.r.t. the divergence parameter $\eta$. This meta-loss function is designed to evaluate the desired properties of the approximate distribution, e.g. test log-likelihood. During meta-testing, a new task is sampled from $p(\mathcal{T})$, and the learned divergence $D_\eta$ is used to optimize the variational distribution.

Besides the above setting, we also consider a few-shot learning set-up similar to the model-agnostic meta-learning (MAML) framework (Finn et al., 2017; 2018; Kim et al., 2018). In this case the probabilistic model architecture is shared by all the tasks, and the goal of meta-training is to obtain a divergence as well as an initialization of the variational parameters $\phi$ for unseen tasks. After meta-training, we will train the model with the learned divergence objective and the learned initialization of variational parameters $\phi$ on new tasks. The above meta-learning settings are practical as demonstrated in many previous work (Finn et al., 2017; 2018; Kim et al., 2018), including meta-learning for Bayesian inference (Gong et al., 2018). Attaining common knowledge based on the previous tasks has been proved to be useful for the future tasks.

**Algorithm 1** Meta-$D$

**Input:** $M$: number of training tasks. $\beta$, $\gamma$: learning rate hyperparameters.
Initialize $\eta$, $\phi_i$, $i = 1, \ldots, M$ ($\phi_i$ can have different structures).
**loop**
   **for** $T_i$, $i = 1, \ldots, M$ **do**
      **for** $B$ times **do**
         Update the variational parameters with the current divergence: $\phi_i \leftarrow \phi_i - \beta \nabla_{\phi_i} D_\eta(q_{\phi_i}, T_i)$.
      **end for**
   **end for**
   Update $\eta \leftarrow \eta - \gamma \nabla_\eta \frac{1}{M} \sum_i \mathcal{J}(q_{\phi_i}, T_i)$
**end loop**
**Output:** $\eta$

**Algorithm 2** Meta-$D \& \phi$

**Input:** $p(\mathcal{T})$: distribution over tasks. $\beta$, $\gamma$, $\tau$: learning rate hyperparameters.
Initialize $\phi$, $\eta$
**loop**
   Sample $M$ tasks $T_i \sim p(\mathcal{T})$.
   **for all** $T_i$ **do**
      Update the variational parameters with the current divergence: $\phi_i \leftarrow \phi - \beta \nabla_\phi D_\eta(q_\phi, T_i)$.
   **end for**
   Update $\phi \leftarrow \phi - \tau \nabla_\phi \frac{1}{M} \sum_i \mathcal{J}(q_{\phi_i}, T_i)$; $\eta \leftarrow \eta - \gamma \nabla_\eta \frac{1}{M} \sum_i \mathcal{J}(q_{\phi_i}, T_i)$
**end loop**
**Output:** $\eta$, $\phi$

## 3.1 META-LEARNING DIVERGENCE OBJECTIVE (META-D)

We consider the first problem setting of learning a divergence. We first present our method assuming the parameteric form of $D_\eta$ is given. Then we will provide the details of parameterization of two divergence families: $\alpha$-divergence and $f$-divergence and how they fit in this framework.

The idea of meta-learning divergences is that we first optimize the approximate posterior by minimizing the current divergence, then update the divergence using the feedback from the meta-loss. Formally speaking, for each task $T_i$ we perform $B$ gradient descent steps on the variational parameters $\phi_i$ using the current divergence $D_\eta$ as in the typical VI optimization:

$$\phi_i \leftarrow \phi_i - \beta \nabla_{\phi_i} D_\eta(q_{\phi_i}, T_i). \tag{6}$$

By doing so the updated variational parameters are a function of the divergence parameters $\eta$. Then we update the divergence parameters $\eta$ by one-step gradient descent using the meta-loss $\mathcal{J}$:

$$\eta \leftarrow \eta - \gamma \nabla_\eta \frac{1}{M} \sum_i \mathcal{J}(q_{\phi_i}, T_i) \tag{7}$$

We call meta learning divergence objective *meta-D* and outline the algorithm in Algorithm 1.

**Meta-learning within $\alpha$-divergence family** The parameterization of Renyi's $\alpha$-divergence (2) is straightforward: $\eta = \alpha$. As the VR bound (3) is an equivalent optimization objective to Renyi's $\alpha$-divegence, it means $\nabla_{\phi_i} D_\eta = -\nabla_{\phi_i} \mathcal{L}_\alpha$, and in practice this gradient is computed using Eq.(4) ($p(\theta, x)$ in Eq.(4) is computed by $p(x|\theta)p(\theta)$). The direct computation of Eq.(4) is beneficial as it is continuous in $\alpha \in (0, +\infty)$. This is in contrast with automatic differentiation of $\alpha$-divergences which has numerical issues due to the dis-continuity of $\alpha$-divergences for certain $\alpha$ values (Li & Turner, 2016; Minka et al., 2005).

**Meta-learning within $f$-divergence family** We wish to parameterize the $f$-divergence (5) by parameterizing the convex function $f$ using a neural network. However, it is less straight-forward to specify the convexity constraint for neural networks . Fortunately, the $f$-divergence and its gradient can be specified through its second order derivative $f''$ without the original $f$ (Wang et al., 2018a).

**Proposition 1.** *Assume $\nabla_\theta \log\left(\frac{p(\theta)}{q_\phi(\theta)}\right)$ exists, then*

$$\nabla_\phi D_f(p||q_\phi) = -\mathbf{E}_{\epsilon, \theta = h_\phi(\epsilon)}\left[ g_f\left(\frac{p(h_\phi(\epsilon))}{q_\phi(h_\phi(\epsilon))}\right) \nabla_\phi h_\phi(\epsilon) \nabla_\theta \log\left(\frac{p(\theta)}{q_\phi(\theta)}\right) \right] \tag{8}$$

*where $g_f(t) = f''(t)t^2$.*

The above proposition implies that we can define the gradient of $f$-divergence through $f''$. The following proposition guarantees that any $g$ is corresponding to a valid $f$.

**Proposition 2.** *For any non-negative function $g$ on $\mathbb{R}_+$, there exists a function $f$ such that $g(t) = f''(t)t^2$. If $g_f(t)$ is strictly positive, i.e. $g_f(1) > 0$, then $D_f(p||q_\phi) = 0$ implies $p = q_\phi$.*

See Wang et al. (2018a) for the proofs. Given these guarantees, we propose to parameterize $f$ implicitly by parameterizing $g_f$ which can be any non-negative function. We turn the problem into using a neural network to express a non-negative function who is strictly positive at $t = 1$. For computational convenience, we further restrict the form of the function to be $g_f(t) = \exp(h_\eta(t))$ where $h_\eta(t)$ is a neural network with parameters $\eta$. This definition of $g_f$ is strictly positive for all $t$, which clearly satisfies the assumption of Proposition 2. Then by using Eq. (8) to compute the gradient $\nabla_{\phi_i} D_\eta = \nabla_{\phi_i} D_{f_\eta}$ (see appendix A for details), it is clear to see that the $f$-divergence is learnable through Algorithm 1.

## 3.2 Meta-Learning Divergence Objective and Variational Parameters

In addition to learning the divergence objective, we also consider the setting where fast adaptation of the variational parameters to new tasks is desirable. Similar to MAML, the probabilistic models $\{p_{T_i}(\theta_i, \mathcal{D}_{T_i})\}$ share the same architecture, and the goal is to learn an initialization of variational parameters $\phi_i \leftarrow \phi$. On a specific task, $\phi$ is adapted to be $\phi_i$ according to the learnable divergence $D_\eta$ (which can be $-\mathcal{L}_\alpha$ or $D_{f_\eta}$).

$$\phi_i \leftarrow \phi - \beta \nabla_\phi D_\eta(q_\phi, T_i) \tag{9}$$

Again the updated $\phi_i$ is a function of both $\eta$ and $\phi$. Here we simply assume the number of gradient steps to be $B = 1$, and it is straightforward to extend the method to $B > 1$. For meta-update, besides updating divergence parameters $\eta$ with Eq.(7), we also use the same meta-loss to update $\phi$:

$$\phi \leftarrow \phi - \tau \nabla_\phi \frac{1}{M} \sum_i \mathcal{J}(q_{\phi_i}, T_i) \tag{10}$$

We call meta-VI with learning both the divergence objective and variational parameters' initialization *meta-D&$\phi$* and summarize the algorithm in Algorithm 2. Similar to the previous section, the divergence families in consideration are $\alpha$-divergences and $f$-divergences.

## 4 Experiments

We evaluate the proposed meta-VI approaches on three tasks: gaussian mixture approximation, Bayesian neural network regression, and recommender system with partial variational auto-encoder. We use marginal log-likelihood as the meta-loss for all experiments except for the Gaussian mixture approximation task where we evaluate the method using a variety of meta-losses to directly demonstrate the ability of our method to learn the optimal divergence.

### 4.1 Approximate Mixture of Gaussians

We first verify the proposed meta-VI approach can learn a good divergence by considering a 1-d distribution approximation problem. Each task includes approximating a mixture of two Gaussians by a Gaussian distribution which is attained by minimizing $\min_\phi D_\eta(q_\phi || p)$. The mixture of Gaussian distribution $p(\theta) = 0.5\mathcal{N}(\theta; \mu_1, \sigma_1^2) + 0.5\mathcal{N}(\theta; \mu_2, \sigma_2^2)$ is generated by

$$\mu_1 \sim \text{Unif}[0, 3] \quad \sigma_1 \sim \text{Unif}[0.5, 1.0]; \qquad \mu_2 = \mu_1 + 3 \quad \sigma_2 = \sigma_1 * 2$$

Therefore each task has a different target distribution but with similar properties (i.e. the distance between two modes is the same and the standard deviation of the second mode is 2 times larger than that of the first mode). The choice of the divergence affects the properties of the approximated Gaussian distribution as shown in Figure 1.

We test the meta-VI approach with two types of meta-loss: $D_{0.5}(q||p)$ and total variation (TV). If $D_{0.5}(q||p)$ is the metric we care when evaluating the quality of the approximation $q$ to the target $p$, then a good divergence will be $D_{0.5}(q||p)$ itself. Therefore the goal of testing with meta-loss $D_{0.5}$ is to verify that our method is able to learn the preferred divergence given a rich enough family of candidate divergences $\{D_\eta\}$. As in this case the preferred divergence is known, we can directly evaluate the learned divergence by comparing it with the known preferred divergence. In practice,

Table 1: Meta-$D$ on MoG: learned value of $\alpha$ from meta-$\alpha$ and BO. BO with 8 iterations has similar running time as meta-$\alpha$.

| Methods | $\alpha = 0.5$ | TV |
|---|---|---|
| meta-$\alpha$ | 0.52±0.01 | 0.31±0.01 |
| BO (8 iters) | 0.81±0.03 | 0.69±0.08 |
| BO (16 iters) | 0.54±0.07 | 0.32±0.03 |

Table 2: Meta-$D$ on MoG: value of meta-loss over 10 test tasks.

| Methods | $\alpha = 0.5$ | TV |
|---|---|---|
| ground truth | 0.0811±0.0277 | - |
| meta-$\alpha$ | 0.0811±0.0277 | 0.0855±0.0149 |
| meta-$f$ | **0.0795**±0.0301 | **0.0806**±0.0163 |
| BO (8 iters) | 0.0833±0.0289 | 0.0879±0.0143 |
| BO (16 iters) | 0.0811±0.0277 | 0.0855±0.0149 |

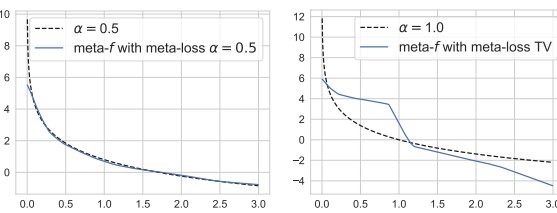

Figure 2: Meta-$D$ on MoG: visualizing the learned $\log f''$.

Table 3: Meta-$D$ on MoG: rank of meta-loss over 10 test tasks.

| Methods | $\alpha = 0.5$ | TV |
|---|---|---|
| meta-$\alpha$ | **2.10**±0.70 | 2.10±0.30 |
| meta-$f$ | **2.10**±1.37 | **1.00**±0.00 |
| BO (8 iters) | 3.50±0.67 | 4.00±0.00 |
| BO (16 iters) | 2.30±0.90 | 2.90±0.30 |

Table 4: Meta-$D\&\phi$ on MoG: learned value of $\alpha$.

| Methods | $\alpha = 0.5$ | TV |
|---|---|---|
| meta-$\alpha\&\phi$ | 0.88 | 0.77 |

the desired evaluation metric for approximation quality (e.g. test log-likelihood) typically does not belong to $\alpha$- or $f$-divergence family; to test this scenario we use TV to evaluate the performance of our method when meta-loss is beyond the divergence family. TV is a common distance measure for probability distributions. It is defined as

$$TV(p, q) = \sup_x |p(x) - q(x)| = \frac{1}{2} \int |p(x) - q(x)| dx.$$

For $\alpha \in (0, 1]$, TV is related to $\alpha$-divergence by $\frac{\alpha}{2} TV^2 \leq D_\alpha(p||q)$ (Gilardoni, 2010).

We first test meta-learning the divergence objective (Algorithm 1). If the divergence set is $\alpha$-divergence which is parameterized by a scalar, then an alternative approach is to treat the divergence learning task as a hyperparameter search problem and use Bayesian optimization (BO) (Snoek et al., 2012) to solve it. Therefore, we use BO as a baseline for meta-$\alpha$. We note that BO is not applicable when the divergence set is $f$-divergence which is parameterized by a neural network. We set the search region for BO to be $\alpha \in [0, 3]$ which includes many common divergence such as KL, Helinger distance ($\alpha = 0.5$) and $\chi^2$-divergence ($\alpha = 2$). We learn the divergence on $M = 10$ tasks and set $B = 1$. BO is used to find the optimal $\alpha$ value for these 10 tasks (see appendix B.2 for more details about BO).

In Table 1, we report the learned value of $\alpha$ from meta-$\alpha$ and BO. When the meta-loss is $D_{0.5}(q||p)$, the learned $\alpha$ from meta-$\alpha$ is very close to 0.5 which demonstrates that our method can essentially learn a good $\alpha$. On the other hand, BO is less computationally efficient, as it needs to train a model from scratch every single time when evaluating a new value of $\alpha$, while our method can update $\alpha$ based on the current model. We also consider learning $f$-divergence and visualize in Figure 2 the learned $\log f''$. When $D_{0.5}(q||p)$ is in use as the meta-loss, the corresponding $\log f^{*''}$ for $D_{0.5}(q||p)$ is analytical, and we see from Figure 2(a) that the learned $\log f''$ and $\log f^{*''} + 0.8$ are almost identical. This means meta-VI has learned the optimal divergence $D_{0.5}(q||p)$ (for any positive $a$, $f(t)$ and $af(t)$ define the same divergence).

In the case of using TV as the meta-loss, the optimal divergence is not analytic. Therefore, we instead report in Table 2 the meta-losses on 10 test tasks, which are obtained by executing the learned divergence minimization algorithm for 2000 iterations. The error bar is large due to the large variance among different tasks, so we also report the ranking in Table 3. It clearly shows that meta-$\alpha$ and meta-$f$ are superior over BO. Moreover, meta-$f$ outperforms meta-$\alpha$ when the meta-loss is TV. From Figure 2 (b), we can see that the learned $f$-divergence is not inside $\alpha$-divergence, showing the benefit of using a larger divergence family.

Next we test meta-learning both the divergence objective and the variational parameters (Algorithm 2). We use Algorithm 2 without updating divergence as a baseline, denoting by VB&$\phi$. During

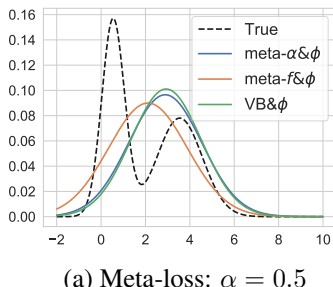 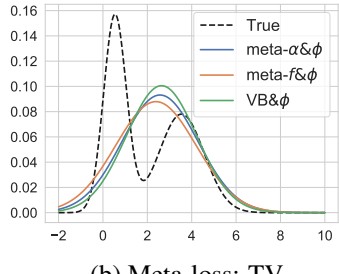

(a) Meta-loss: $\alpha = 0.5$        (b) Meta-loss: TV

Figure 3: Meta-$D\&\phi$ on MoG: visualization of approximate distribution after 20 updates.

training, we sample 10 tasks each time and perform $B = 20$ inner loop gradient update. The learned $\alpha$ is different from Table 1 (see Table 4). We conjecture that this is related to the learned $\phi$ and the horizon length. During meta-testing, we use the learned $\phi$ for variational parameter initialization, and train the variational parameters with the learned divergence for 20 and 100 iterations respectively to evaluate the effect of the learned divergence in short and long horizon. We summarize the meta-loss in Table 5 and the ranking in Table 6. Our method are not only better than VB$\&\phi$ after 20 updates but also better after 100 updates. This demonstrate the benefit of learning a divergence for the tasks instead of the conventional VB. To further explore the reason of getting lower meta-loss of meta-$D\&\phi$, we visualize the approximate distribution of all methods after 20 steps in Figure 3. The approximate distributions obtained by meta-$D\&\phi$ tends to fit the mixture of Gaussians more globally (mass-covering) than VB$\&\phi$. This mass-covering behaviour result in better meta-loss. Compared to learning divergence only, learning variational parameter initialization helps shorten the training time on new tasks (100 iterations v.s. 2000 iterations). Notably, meta-VI is able to provide this initialization along with divergence learning without extra cost.

Table 5: Meta-$D\&\phi$ on MoG: value of meta-loss over 10 test tasks.

| Methods\Meta-loss | $\alpha = 0.5$ (20 iters) | TV (20 iters) | $\alpha = 0.5$ (100 iters) | TV (100 iters) |
|---|---|---|---|---|
| meta-$\alpha\&\phi$ | $0.1207 \pm 0.0500$ | $0.0982 \pm 0.0166$ | $0.0879 \pm 0.0305$ | $\mathbf{0.0903} \pm 0.0149$ |
| meta-$f\&\phi$ | $\mathbf{0.0793} \pm 0.0237$ | $\mathbf{0.0935} \pm 0.0152$ | $\mathbf{0.0784} \pm 0.0332$ | $0.0918 \pm 0.0151$ |
| VB$\&\phi$ | $0.1237 \pm 0.0539$ | $0.1026 \pm 0.0181$ | $0.0905 \pm 0.0332$ | $0.0926 \pm 0.0153$ |

Table 6: Meta-$D\&\phi$ on MoG: rank of meta-loss over 10 test tasks.

| Methods\Meta-loss | $\alpha = 0.5$ (20 iters) | TV (20 iters) | $\alpha = 0.5$ (100 iters) | TV (100 iters) |
|---|---|---|---|---|
| meta-$\alpha\&\phi$ | $2.10 \pm 0.54$ | $1.80 \pm 0.60$ | $2.20 \pm 0.75$ | $\mathbf{1.40} \pm 0.66$ |
| meta-$f\&\phi$ | $\mathbf{1.20} \pm 0.60$ | $\mathbf{1.50} \pm 0.81$ | $\mathbf{1.40} \pm 0.80$ | $2.10 \pm 0.83$ |
| VB-MAML | $2.70 \pm 0.46$ | $2.70 \pm 0.46$ | $2.40 \pm 0.49$ | $2.50 \pm 0.50$ |

## 4.2 REGRESSION TASKS WITH BAYESIAN NEURAL NETWORKS

The second test considers Bayesian neural network regression. The distribution of ground truth regression function is defined by a sinusoid function with heteroskedastic noise (which is a function of $x$, see Figure 4 (a)): $y = A\sin(x+b) + A/2|\cos((x+b)/2)|\epsilon$, where the amplitude $A \in [5, 10]$, the phase $b \in [0, 1]$ and $\epsilon \sim \mathcal{N}(0, 1)$. The heteroskedastic noise makes the uncertainty estimate crucial when compared with the sinusoid function fitting task in Finn et al. (2017); Kim et al. (2018). The model is a two-layer neural network with hidden layer size 20 and RELU nonlinearities.

For meta-learning divergence only, the training set size is 1000 and is obtained by sampling $x \in [-4, 4]$ uniformly. We use $M = 20$, $B = 1$, $K = 50$ and batch size 40 of which 20 data points are for updating $\phi_i$ Eq.(6) and 20 points are for updating $\eta$ Eq.(7). We train meta-$D$ for 1500 epochs. To evaluate the performance, we train the model with the learned divergence and VB respectively on new tasks for 1000 epochs. The quantitative results are summarized in Table 7. We can see that the test log-likelihood of both meta-$\alpha$ and meta-$f$ are significantly better than VB and the root mean square error (RMSE) are similar for all methods. We visualize the predictive distribution on an example sinusoid function in Figure 4. All methods fit the mean well which is consistent with the RMSE results. However, VB fails to capture the heteroskedastic uncertainty and instead used

Table 7: Meta-$D$ on Sin: results are over 10 test tasks (1000 epochs).

|  | Test LL | RMSE |
|---|---|---|
| VB | -0.6377±0.0433 | 0.4522±0.0196 |
| meta-$\alpha$ | -0.4596±0.0857 | 0.4500±0.0236 |
| meta-$f$ | **-0.4390**±0.1084 | 0.4599±0.0200 |

Table 8: Meta-$D\&\phi$ on Sin: results are over 10 test tasks (500 epochs).

|  | Test LL | RMSE |
|---|---|---|
| VB&$\phi$ | -0.6354±0.0599 | 0.4556±0.0247 |
| meta-$\alpha\&\phi$ | -0.4967±0.0647 | 0.4562±0.0207 |
| meta-$f\&\phi$ | **-0.4852**±0.0853 | 0.4552±0.0217 |

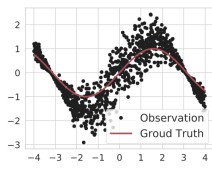 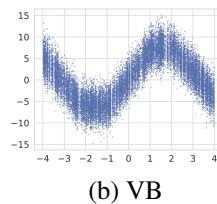 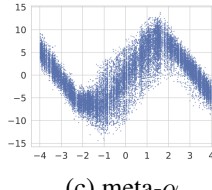 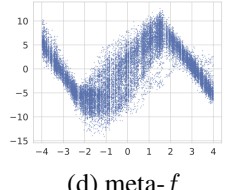

(a) Ground Truth      (b) VB      (c) meta-$\alpha$      (d) meta-$f$

Figure 4: Meta-$D$ on sin: the predictive distribution on a sinusoid wave.

homoskedastic noise to fit the data. On the other hand, meta-$\alpha$ and meta-$f$ can reason about the heteroskedastic noise. This explains the results of better test log-likelihood.

For learning both divergence and variational parameters initialization, we sample 20 tasks where each task has 40 data points. We use 20 points for $\phi_i$ Eq.(9) and the other 20 points for updating divergence $\eta$ Eq.(7) and the shared initialization $\phi$ Eq.(10). We set $B = 1$. To evaluate, we start with the learned initialization and train the variational parameters with the learned divergence for 500 epochs. Similar to the results of learning only the divergence objective, meta-$\alpha\&\phi$ and meta-$f\&\phi$ are able to model heteroskedastic predictive distribution while VB&$\phi$ cannot. The quantitative evaluation are given in Table 8 and an example of predictive distribution is given in Figure 6 (see appendix). Meta-$D\&\phi$ converges faster than meta-$D$, indicating that learning model initialization can shorten the training time on new tasks.

### 4.3 RECOMMENDER SYSTEM WITH PARTIAL VARIATIONAL AUTO-ENCODERS

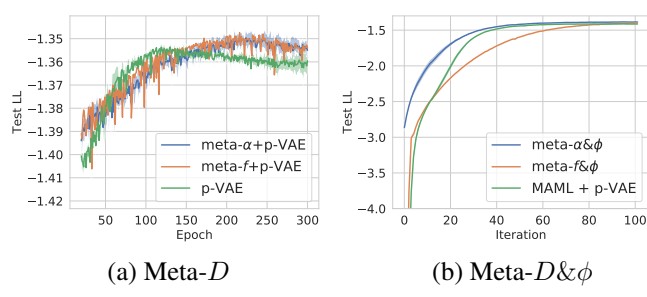

(a) Meta-$D$      (b) Meta-$D\&\phi$

Figure 5: Test log-likelihood of meta-VI on MovieLens. (b) The final results of meta-$\alpha\&\phi$, meta-$f\&\phi$ and MAML+p-VAE are -1.3855, -1.3985 and -1.4140 respectively.

We test our method on recommender systems with Partial Variational Auto-encoders (p-VAEs). P-VAE is a recently proposed model to deal with partially observed data and has been used to do user rating prediction in recommender system (Ma et al., 2018b;a). Similar to vanilla VAE (Kingma & Welling, 2013), p-VAE uses the KL-divergence as the variational objective. We apply our proposed method to the divergence objective in p-VAE.

We consider MovieLens 1M dataset (Harper & Konstan, 2016) which contains 1,000,206 ratings of 3,952 movies from 6,040 users. We split the users into seven age groups: under 18, 18-24, 25-34, 35-44, 45-49, 50-55 and above 56, and regard predicting the ratings of users within the same age group as a task since the users with similar age may have similar preferences. We select four as training tasks (under 18, 25-34, 45-49, above 56) and use the remaining as test tasks.

For the setting of learning divergence only, during meta-training, we sample 100 users per task (400 users in total) and use half of the observed ratings to compute Eq.(6) and the other half for computing the meta-loss. The number of training epochs is 400. During meta-testing, we use 90%/10% training-test split for the three test tasks and train p-VAE with the learned divergence. The baseline p-VAE is directly trained on test tasks with KL-divergence. From Figure 5 (a), we can see that the combination of meta-$D$ and p-VAE outperforms vanilla p-VAE in terms of test log-likelihood, showing that meta-$D$ has learn a suitable divergence that leads to better test performance.

For learning both divergence and variational parameters, the setup of training is the same as learning divergence only except that now we also perform updates in Eq.(9). We compare our method with getting a p-VAE model initialization only (obtained by Algorithm 2 without updating $\eta$). This can be regarded as a combination of MAML and p-VAE. During evaluation, we apply 60%/40% training-test split for the test tasks and train the learned p-VAE model with learned divergence. Figure 5 (b) implies that all methods can converge quickly on the new task with only 100 iterations. Both meta-$\alpha \& \phi$ and meta-$f \& \phi$ are better than p-VAE at the beginning, indicating that the learned divergence can help fast adaptation. Besides, meta-$\alpha \& \phi$ and meta-$f \& \phi$ also converge better than p-VAE in the end. This shows the learned divergence helps in both short and long horizon.

## 5 RELATED WORK

**Variational Inference**    Variational inference (VI) has advanced rapidly in recent years (Zhang et al., 2018). Several works have introduced a new divergence family for VI (Li & Turner, 2016; Hernández-Lobato et al., 2016; Bamler et al., 2017). Another line of work improve the importance sampling estimate of model evidence by increasing the number of importance sampling particles (Burda et al., 2015) or increasing the signal-to-noise ratio of gradient estimate (Rainforth et al., 2018). Stochastic optimization methods have also been deployed to scale up VI (Hoffman et al., 2013; Li et al., 2015; Dehaene & Barthelmé, 2018). Our work is related to the work that improves the variational objective with alternative divergence measures; the difference is that our divergence measure is learnable and can be selected in an automatic fashion for a certain type of tasks.

**Meta-Learning/few-shot learning**    Recent work has applied Bayesian modelling techniques to enhance uncertainty estimate for meta-learning/few-shot learning (Grant et al., 2018; Finn et al., 2018; Kim et al., 2018; Ravi & Beatson, 2018). They regard the framework of MAML (Finn et al., 2017) as hierarchical Bayes and conduct Bayesian inference on meta-parameters and/or task-specific parameters. Grant et al. (2018); Kim et al. (2018) applied Bayesian inference to task-specific parameters with Laplacian approximation and Stein variational gradient descent, respectively. Finn et al. (2018) instead approximated the exact posterior over meta-parameters using variational inference but still kept point estimate for task-specific parameters. Ravi & Beatson (2018) obtain posteriors over both meta-parameters and task-specific parameters with variational inference. Our focus is distinct from this line work in that our research is the opposite direction: leveraging the idea of meta-learning to advance Bayesian inference. Additionally, our meta-$D \& \phi$ without learning divergence can be regarded as a different Bayesian MAML method: we do not follow hierarchical Bayes but directly train variational parameters so that it can quickly adapt to new tasks.

**Meta-Learning for loss functions**    Our meta-learning method is also related to meta-learning a loss function. In reinforcement learning, Houthooft et al. (2018) meta-learned the loss function for updating policy. Xu et al. (2018) meta-learned the value function to interact with the environment. Our work extends the idea of a learnable loss function to Bayesian inference.

**Meta-Learning for Bayesian inference algorithms**    A recent attempt to meta-learning stochastic gradient MCMC (SG-MCMC) is presented by Gong et al. (2018), which proposed to meta-learn the diffusion and curl matrices of the SG-MCMC's underlying stochastic differential equation. Also Wang et al. (2018b) applied meta-learning to build efficient and generalizable block-Gibbs sampling proposals. Our work is distinct from previous work in that we apply meta-learning to improve VI, which is a more scalable inference method than MCMC. To the best our knowledge, we are the first to study the automatic choice and design of VI inference algorithms.

## 6 CONCLUSION

We propose meta-VI which automates the choice of divergence objective in VI via meta-learning. It further allows meta-learning of variational parameter initialization for fast adaptation on new tasks. Within the framework of meta-VI, we consider two divergence families, $\alpha$-divergence and $f$-divergence, and design parameterizations of divergences to enable learning via gradient descent. Experimental results on Gaussian mixture approximation, regression with Bayesian neural networks and recommender systems demonstrate the improvement of meta-VI over vanilla VI, which shows the benefits of learning a suitable divergence measure tailored to the specific tasks at hand.

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

## A    COMPUTING EQUATION (8) IN PRACTICE

With dataset $\mathcal{D}$, the density ratio in f-divergence becomes $\frac{p(\theta|D)}{q_\phi(\theta)} = \frac{p(D|\theta)p(\theta)}{q_\phi(\theta)p(D)}$. We estimate $p(D)$ through importance sampling and MC approximation: $p(D) = E_{\theta \sim p(\theta)}[p(D|\theta)] = E_{\theta \sim q_\phi(\theta)}[\frac{p(D|\theta)p(\theta)}{q_\phi(\theta)}] \approx \frac{1}{K} \sum_k^K \frac{p(D|\theta_k)p(\theta_k)}{q_\phi(\theta_k)}$ where $\theta_k \sim q_\phi(\theta)$. After doing this, the density ratio becomes $\frac{p(\theta_k|D)}{q_\phi(\theta_k)} = \frac{p(D|\theta_k)p(\theta_k)}{q_\phi(\theta_k)} \Big/ \frac{1}{K} \sum_k^K \frac{p(D|\theta_k)p(\theta_k)}{q_\phi(\theta_k)}$ which can be regarded as a self-normalized estimator, similar to the normalization importance weight in Li & Turner (2016). A self-normalized estimator generally helps stabilize the training especially at the beginning. We use this estimator for regression tasks and recommender system.

## B    ADDITIONAL EXPERIMENTAL RESULTS AND SETTING DETAILS

### B.1    MODEL ARCHITECTURE FOR $f$-DIVERGENCE

On all experiments, we parameterize $g(t)$ in $f$-divergence by a neural network with 2 hidden layers with 100 hidden units and RELU nonlinearilities.

### B.2    APPROXIMATE MIXTURE OF GAUSSIANS

The expectation in Eq.(3) and (8) is computed by MC approximation with 1000 particles. Note that $p(\theta)$ is computable, since we know the parameters of $p$.

Bayesian optimization is implemented through a public package.[1] The acquisition function is the upper confidence bound with kappa 0.1. We used the same data of the training tasks for BO. Specifically, the objective function that BO wants to minimize is the meta-loss ($D_{0.5}$ or TV). Every time BO selects an $\alpha$, we train 10 models with that $\alpha$-divergence on the support sets of 10 training tasks respectively and get the mean of log-likelihood on the query sets of the 10 training tasks. Each time the model is trained for 2000 iterations.

When $D_{0.5}$ is in use as the meta-loss, ideally the learned f-divergence should be close to $D_{0.5}$. When the f-divergence is $D_{0.5}$, the $f$ function is $f(t) = \frac{t^{0.5}}{-0.5^2}$, and the analytical expression of $\log f''(t)$ is $-1.5 \log t + C$ with $C$ reflecting the scaling constant in $f$. In Figure 2, we compare the learned $\log f''(t)$ and the ground truth $-1.5 \log t + C$. We found that the learned $\log f''(t)$ is very close to $-1.5 \log t + 0.8$. This means that our method has learned the optimal divergence $D_{0.5}$ (because the definition of $f$-divergence is invariant to constant scaling of the function $f$, i.e. $f$ and $e^{0.8} \times f$ define the same divergence).

### B.3    REGRESSION TASKS WITH BAYESIAN NEURAL NETWORKS

We provide the learned value of $\alpha$ from meta-$\alpha$ and meta-$\alpha\&\phi$ in Table 9. The predictive distributions on an example test task are given in Fig. 6. Similar to the results of meta-$D$, meta-$D\&\phi$ is also able to model heteroskedastic noise while VB$\&\phi$ cannot. We have tried BO on this task and the later recommender system task but it appears to be much more inefficient than our methods. Given the similar runtime as our methods, BO can only conduct two search resulting in bad value of $\alpha$. The results of BO are worse than KL-divergence, therefore ignored in the paper.

Table 9: Learned value of $\alpha$ of meta-$\alpha$ and meta-$\alpha\&\phi$ on sinusoid regression.

|  | meta-$\alpha$ | meta-$\alpha\&\phi$ |
|---|---|---|
| $\alpha$ | 0.1666 | 0.1020 |

Table 10: Learned value of $\alpha$ of meta-$\alpha$ and meta-$\alpha\&\phi$ on MovieLens.

|  | meta-$\alpha$ | meta-$\alpha\&\phi$ |
|---|---|---|
| $\alpha$ | 0.9029 | 1.0602 |

---

[1] https://github.com/fmfn/BayesianOptimization

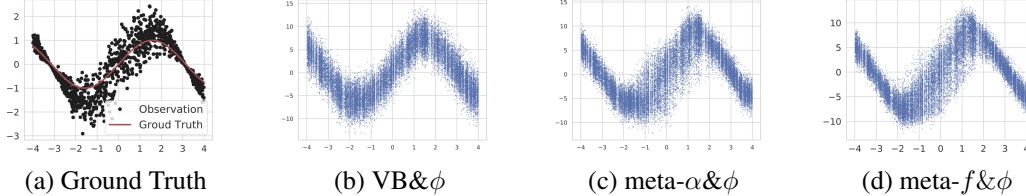

Figure 6: Meta-$D\&\phi$ on sin: Predictive distribution on a sinusoid wave.

## B.4  RECOMMENDER SYSTEM

Again we provide the value of learned $\alpha$ from meta-$\alpha$ and meta-$\alpha\&\phi$ in Table 10. Besides the test log-likelihood, there are other popular evaluation metric being used in recommender system and sometimes they are not consistent with each other. Therefore, we also evaluate the performance of our method in terms of other common metrics: test root mean square error (RMSE) and test mean absolute error (MAE). For both metrics, our methods converge better than the baseline in the setting of learning inference algorithm and the setting of learning inference algorithm and model parameters (see Figure 7 and 8).

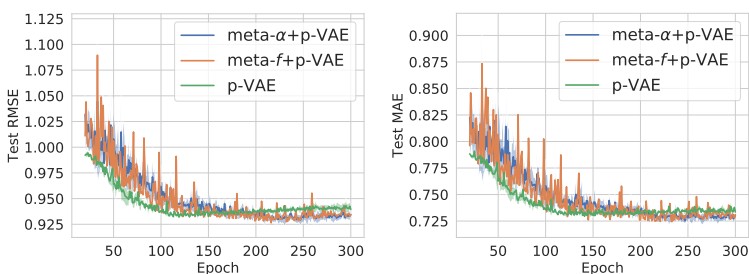

Figure 7: Meta-$D$ on ML: Comparison of meta-$D$ and p-VAE in terms of test RMSE and test MAE.

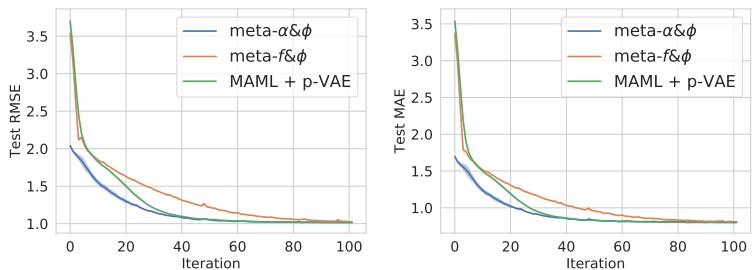

Figure 8: Meta-$D\&\phi$ on ML: Comparison of meta-$D\&\phi$ and MAML+p-VAE in terms of test RMSE and test MAE.

## C  VAE ON MNIST

We added an additional experiment on the real-word dataset, MNIST, to further demonstrate the effectiveness of our methods. Specifically, we trained a variational auto-encoder (VAE) on MNIST and replaced the KL divergence in VAE by our learned $\alpha$-divergence. Each digit is regarded as a task and we let the first 5 digits (0-4) to be the training tasks and the last 5 digits (5-9) to be the test tasks. For learning divergence only, during meta-training, we sample 128 images each task and use half of the images to compute Eq.(6) and the other half for computing the meta-loss. The number of training epochs is 600. During meta-testing, we train VAE with the learned divergence on the training set of the test tasks for 300 epochs and compute the marginal log-likelihood on the test set of

Table 11: Meta-$\alpha$ on MNIST: marginal log-likelihood on 5 test tasks.

| Digit | 5 | 6 | 7 | 8 | 9 |
|---|---|---|---|---|---|
| VAE | -133.94 | -121.74 | -92.20 | -145.32 | -120.55 |
| meta-$\alpha$+VAE | -133.59 | -120.86 | -92.06 | -144.84 | -120.24 |

Table 12: Meta-$\alpha\&\phi$ on MNIST: marginal log-likelihood on 5 test tasks.

| Digit | 5 | 6 | 7 | 8 | 9 |
|---|---|---|---|---|---|
| VAE + MAML | -139.06 | -129.56 | -99.76 | -149.76 | -124.91 |
| meta-$\alpha\&\phi$ | -134.75 | -124.35 | -92.71 | -145.99 | -119.74 |

Table 13: Learned value of $\alpha$ of meta-$\alpha$ and meta-$\alpha\&\phi$ on MNIST.

| | meta-$\alpha$ | meta-$\alpha\&\phi$ |
|---|---|---|
| $\alpha$ | 0.15 | 0.97 |

the test tasks. $B = 1$ and $K = 10$. The baseline VAE means the standard VAE with KL-divergence. Similar to the previous tasks, we have considered BO on this task but found that it is very inefficient. For meta-learning both divergence and the model initialization, the setup of training is the same as learning divergence except that now all tasks share the same model initialization. We compare our method with getting a VAE model initialization only (obtained by Algorithm 2 without updating $\eta$). This can be regarded as a combination of MAML and VAE. During evaluation, we train the VAE with the learned divergence and the model initialization on the training set of the test tasks for 200 epochs and evaluate the marginal log-likelihood on the test set.

We use the same architecture (100 hidden units and 3 latent variables) and the marginal log-likelihood estimator as in Kingma & Welling (2013).

We report the test marginal log-likelihood for each test digit in Table 11 and 12. Overall, these results align with other experiments that the meta-$\alpha$ and meta-$\alpha\&\phi$ are both better than their counterparts. Meta-$\alpha$ and meta-$\alpha\&\phi$ are better than the vanilla VAE on all 5 test tasks, indicating our methods have learned a suitable divergence.

