# OpenReview forum: "Meta-Learning for Variational Inference"
_ICLR.cc/2020/Conference — Reject_

### Official Review · AnonReviewer2 · 2019-10-17
**Official Blind Review #2**

**Rating:** 3

**Review:**

This paper proposes to use a meta-learning approach to learn the divergence used in variational inference and initial variational parameters. It considers two families of learnable divergences, including the alpha-divergence and f-divergence, and proposes two double loop algorithms. In both algorithms, the inner loop adjusts the variational parameters for a specific variational inference task. The outer loop adjusts the parameters in the divergence (and perhaps an initial variational parameter for all tasks) in terms of a human-designed meta-loss. Proposition 1 (following Wang et al. 2018a) shows that the gradients with respect to the meta-parameters in the f-divergence can be obtained using f''. This makes it convenient to parameterize f'' by neural networks to implicitly model f.

The experiments show that the proposed method can outperform a Bayesian optimization (BO) baseline if the meta-loss is the metric of interest in the MoG example. For the regression and recommendation system examples, it compares with a VB baseline and a p-VAE baseline and shows better results.

As for the significance of the paper, I have the following issues:

1. To apply the proposed method, we need to have a family of tasks that shares similarity at hand, which seems to be restricted. For instance, the first two examples are synthesized by hand. The recommendation system example is from real life but also crafted by splitting groups by age. If we just use all of the data by a p-VAE and fine-tune the hyperparameters a little bit (meta-learning needs more time than a p-VAE in a single configuration), can we obtain better results? If so, the last example is not appealing.

2. To apply the proposed method, we need to have some knowledge about our preference in the evaluation and encode that knowledge into the meta-loss. In the first example, the comparison with BO in the alpha=.5 case is unfair because Meta-D uses this knowledge as the meta-loss while BO does not leverage such information. Besides, if we already know the preferred divergence, it is not necessary to learn that divergence. For the other settings where Meta-D does not leverage any knowledge about the evaluation preference, it lacks stronger baselines such as BO.

3. The idea is straightforward and the most challenging part of how to model a convex function by a neural network is solved by existing work.

Based on these issues, I think the contribution of the paper is not sufficient and I tend to reject the paper. Also, note that the paper length extends 8 pages.

The motivation should be strengthened and the experiments should be more precise to improve the paper.

**Experience Assessment:**

I have published one or two papers in this area.

**Review Assessment: Checking Correctness Of Derivations And Theory:**

I assessed the sensibility of the derivations and theory.

**Review Assessment: Checking Correctness Of Experiments:**

I assessed the sensibility of the experiments.

**Review Assessment: Thoroughness In Paper Reading:**

I read the paper at least twice and used my best judgement in assessing the paper.

---

> ### Author Response · Authors · 2019-11-15
> **Response to Reviewer #2 (part 1)**
>
> Thank you for your thoughtful review. Our responses are below.
>
> Q1: To apply the proposed method, we need to have a family of tasks that shares similarity at hand, which seems to be restricted.
>
> A1: The setting in the paper follows a typical meta-learning setting [1,5]: we have several training tasks and we train a learner on them to gain common knowledge which will generalize well on the future tasks. The meta-learning setting is in need in many real-world applications where one type of machine learning solution is provided for different tasks, for example, one recommender solution provider may have many customers from different companies or different departments.
>
> In the same setting as a generic meta-learning task, we propose to meta-learn the inference algorithm. The learner in our method is either the divergence (meta-D) or the divergence and the model initialization (meta-D&phi). This meta-learning setting is practical as demonstrated in many previous work [5, 6, 7], including meta-learning for Bayesian inference [1]. Therefore the situation of applying our methods is essentially common in practice.
>
> Q2: For instance, the first two examples are synthesized by hand. The recommendation system example is from real life but also crafted by splitting groups by age. If we just use all of the data by a p-VAE and fine-tune the hyperparameters a little bit (meta-learning needs more time than a p-VAE in a single configuration), can we obtain better results? If so, the last example is not appealing.
>
> A2: The experiments in this paper follow the meta-learning setup as stated in Section 3, which is consistent with the experimental setup in meta-learning research [1,5,6,7] with the aim to reflect the real-world application needs. The mixture of Gaussian experiment is to illustrate the basic principles of our method and demonstrate that our methods are able to learn a good divergence. The sinusoid regression is a benchmark task in meta-learning [5,6,7] and is an application of Bayesian neural networks. The recommender system task aligns with real-world applications where many datasets/tasks from different companies are available for a shared machine learning platform.
>
> Training p-VAE on all of the data is not practical. It means that whenever we get a new task, we have to train the model on all the data including the previous tasks and the new task. What we do in the paper is to follow the meta-learning setting, i.e. we learn the common knowledge based on the tasks we already have. When a new task comes, we can use this knowledge to better deal with this task without training on old tasks. This is clearly more efficient and practical than the former way. Splitting users by age is one set up to simulate this realistic situation. What we care about is to acquire common knowledge of this type of tasks that can generalize well for future tasks, rather than getting good results on the tasks we have.
>
> We additionally provide the experimental results on another real-world dataset, MNIST. Please see the response “Changes in the Revised Paper” for details.
>
> Q3: In the first example, the comparison with BO in the alpha=.5 case is unfair because Meta-D uses this knowledge as the meta-loss while BO does not leverage such information.
>
> A3: BO uses the same information as meta-VI and the comparisons in the paper are fair. The meta-loss is the objective function that BO aims to minimize thus BO also utilizes the knowledge about our preference. Every time BO selects an $\alpha$, we train 10 models with that $alpha$-divergence on the support sets of 10 training tasks respectively and get the mean of $D_{0.5}$ (the meta-loss) on the query sets of the 10 training tasks. Based on the mean of the meta-loss, BO updates the Gaussian process and selects the next $\alpha$.
>
> Q4: If we already know the preferred divergence, it is not necessary to learn that divergence.
>
> A4: We do not know the preferred divergence in practice. Setting the meta-loss to be $D_{0.5}$ is a synthetic setting serving for evaluation. In this setting it allows us to directly evaluate the learned divergence by comparing it with the known preferred divergence and thus can directly verify whether our method is able to learn a good divergence or not. We have shown in the paper that the learned divergence from our methods with $\alpha$-divergence or f-divergence, is indeed very close to the known preferred divergence which demonstrates the effectiveness of our methods. We have also shown the results for other meta-losses (e.g. TV and test log-likelihood) in the paper. In real-life applications, the choice of the meta loss is determined by the application needs.

---

> > ### Author Response · Authors · 2019-11-15
> > **Response to Reviewer #2 (part 2)**
> >
> > Q5: For the other settings where Meta-D does not leverage any knowledge about the evaluation preference, it lacks stronger baselines such as BO.
> >
> > A5: We have tried BO on other settings (regression task and recommender system) but it appears to be much less efficient than our methods. Given the similar runtime as our methods, on the regression and recommender system tasks, BO can only conduct two searches resulting in a bad value of $\alpha$. This is because BO needs to train a model from scratch every single time when evaluating a new value of $\alpha$, while our method can update $\alpha$ based on the current model. The results of BO are even worse than KL-divergence, therefore such a baseline was not used in the paper as it is not practical. We added this discussion in Appendix B.3 in the paper.
> >
> > Q6: The idea is straightforward and the most challenging part of how to model a convex function by a neural network is solved by existing work.
> >
> > A6: We would like to emphasize the significance and novelty of our methods.
> >
> > First, to the best of our knowledge, we are the first to do automated selection and design for the divergence objective in VI by meta-learning. This work is situated within rich literature including meta-learning for Bayesian inference [1], meta-learning for a loss function [2, 3] and modern VI algorithms.
> >
> > Second, it is not trivial to apply meta-learning to VI. The reasons are:
> >
> > i) The standard meta-learning method cannot be used to learn the loss function in the inner loop which is exactly the situation for learning divergence in VI. To solve this problem, the key insight we have is that the updated variational parameters are essentially dependant on the inner loop loss function. We encode our preference into another loss function called the meta-loss to evaluate the updated variational parameters. Thus when we take the gradient of the meta-loss, the gradient is able to flow back to the parameters of inner loss function through the updated variational parameters. By doing this, we enable the learning of the inner loss function which is the divergence in our case.
> >
> > ii) To make $\alpha$-divergence learnable by meta-learning discussed above, we need to parameterize it such that the inner-loop updates are continuous in $\alpha$. It means a naive solution which relies on automatic differentiation of existing $\alpha$-divergences (such as Amari’s, Tsallis’s and Renyi’s $\alpha$-divergence) will fail, due to the fact that these $\alpha$-divergences are not continuous everywhere. Instead, we manually compute the gradient of Renyi’s $\alpha$-divergence which is continuous in $\alpha$, and it is our contention to use it as a way to parameterize $\alpha$-divergence in meta-learning.
> >
> > iii) To make f-divergence learnable, we used a novel parameterization such that it can be parameterized by a standard neural network and optimized by an optimizer like SGD or ADAM. This is different from the existing work [4] in the way that we use deep neural networks parameterization and enable learning the f-divergence through optimization.
> >
> > The divergence plays an important role in VI and is usually difficult to choose in practice despite the various divergences in the literature. This work introduces a principled way to select and design a suitable divergence objective in VI, which enables us to really use the flexibility of a rich divergence family. Therefore we think this paper is timely and makes a sufficient contribution to practical variational inference.
> >
> > [1] Gong et.al., Meta-Learning for Stochastic Gradient MCMC
> > [2] Houthooft et.al., Evolved policy gradients
> > [3] Xu et.al., Meta-gradient reinforcement learning
> > [4] Wang et.al., Variational inference with tail-adaptive f-divergence
> > [5] Finn et.al., Model-Agnostic Meta-Learning for Fast Adaptation of Deep Networks
> > [6] Finn et.al., Probabilistic Model-Agnostic Meta-Learning
> > [7] Kim et.al., Bayesian Model-Agnostic Meta-Learning

---

### Official Review · AnonReviewer3 · 2019-10-23
**Official Blind Review #3**

**Rating:** 8

**Review:**

SUMMARY OF THE PAPER:
This paper proposes a way to automatically select a divergence to use in variational inference (VI) given a set of datasets (tasks). They consider searching within the alpha-
divergence and f-divergence families. The proposed algorithm works as follows: do a few gradient descent steps on the variational parameters given a fixed divergence parameter, then update the divergence parameter by taking the gradient with respect to a meta loss which is a task-specific measure of goodness of the variational distribution (like test log marginal likelihood). The latter gradient is computed through the gradient descent computation in the inner loop.
The second proposed algorithm does the same, but also learns a good initialization for the variational parameters. This initialization is good in the sense that taking one (or a few) gradient descent step on the variational parameters should give us good variational parameters (MAML style). This is done by taking a gradient with respect to this good initialization parameter through the inner loop gradient descent.

STRUCTURE:
The paper is well-written and easy to understand.

NOVELTY:
As far as I know, learning a divergence for VI using meta-learning is new. The related work is discussed well in section 5.

EXPERIMENTS:
There are experiments on three tasks of increasing complexity. In the simpler experiments, careful ablation studies are done. The results generally show that the proposed method is preferable to alternatives.

CONCLUSION:
I recommend acceptance.

**Experience Assessment:**

I have read many papers in this area.

**Review Assessment: Checking Correctness Of Derivations And Theory:**

I assessed the sensibility of the derivations and theory.

**Review Assessment: Checking Correctness Of Experiments:**

I assessed the sensibility of the experiments.

**Review Assessment: Thoroughness In Paper Reading:**

I read the paper thoroughly.

---

> ### Author Response · Authors · 2019-11-15
> **Response to Reviewer #3**
>
> Thank you for your supportive and valuable review. Your understanding of the paper is correct and we appreciate your recognition of the novelty of our work.

---

### Official Review · AnonReviewer1 · 2019-10-27
**Official Blind Review #1**

**Rating:** 3

**Review:**

Summary of the paper:
This paper proposes to meta-learn a parametric divergence measure for variational inference (VI). Specifically, the paper focuses on alpha-divergence and f-divergence, aims at choosing a good alpha or f for a particular task leveraging the experience learned from previous tasks. The meta-learned divergence is applied to approximate a mixture of two Gaussians, regress sinusoid function with Bayesian neural networks, and learn recommender systems with partial VAE.

Summary of my opinion:
I am leaning towards reject this paper because
1) I'm not convinced by their motivation -- "this line of work has also shown that the optimal divergence can vary depending on tasks". There are many key factors to achieve good performance in VI, such as a good likelihood model, a good variational posterior, a good optimizer and so on. The divergence itself, in my opinion, is less significant. If you really care about the flexibility, the Wasserstein distance may be a better choice.
2) Choosing the alpha or f is a hyper-parameter search problem to me. What you need to do is to prepare a validation set and select the best hyper-parameters based on the performance there. It is impractical to collect M tasks and search for hyper-parameters according to some meta-losses.
3) The experimental part is weak -- two toy problems plus an unusual recommendation system problem is not the typical way that people choose to evaluate VI methods. I suggest to evaluate on standard VAE/BNN benchmarks.

Major comments:
1. From eq.(4) and eq.(8), it seems one has to compute the value of the density function p(theta, D) in order to compute the gradients, which is obviously not possible for BNNs and VAEs. More details on how to perform VI with alpha-divergence and f-divergence should be covered.
2. Can you let the meta-loss to be equal to D_eta? Meta-learning algorithms often optimize the same loss function in the inner and outer level. I couldn't see the point why you want to learn a f-divergence while setting the meta-loss to be D_0.5.
3. Do you have a validation set for the Bayesian optimization (BO) baseline or do you use cross-validation? The details of this baseline should be elaborated. In fact, it is possible to choose the best alpha for each task using BO. Should this be compared?
4. What does VB mean in Section 4.2? If VB uses KL rather than alpha-divergence or f-divergence, the outperformance may only suggest that KL is insufficient there.
5. Have you considered input convex neural networks (Amos et al. 2016) for implementing f?

Minor comments:
1. The proofs of Prop 1 & 2 are missing.
2. "However, using this KL divergence for VI has been criticized for under-estimating the uncertainty..." Any reference to this?
3. Figure 1: alpha is used before it is defined.
4. "Other existing definitions of alpha-divergences have dis-continuous gradients at the alpha values corresponding to KL divergences" -- missing reference.
5. "When D_0.5 is in use as the meta-loss, the corresponding log f*'' is analytical" -- could you elaborate on this?

------------------------------
After rebuttal:
I would like to thank the authors for their extremely detailed responses! The paper is now greatly improved.
I would like to increase my score from 3 to 5, however, there is no such option this year.
The reason why I am still hesitated to accept this paper is mainly because I am not sure how useful learning the divergence to VI is. For unsupervised learning, people use GAN or VQ-VAE to obtain realistic samples, but which is not attributed to better divergence measures. Plus, what we refer to "better" depends on the supplied meta-loss, which may not be better at all. Another unsatisfactory point of this paper in my opinion is that it only demonstrated results on toy datasets and MNIST. Although the proposed method is valid for alpha-divergence and f-divergence, I suggest the authors to find a real application which demonstrates the merit of the idea in practice.

**Experience Assessment:**

I have published one or two papers in this area.

**Review Assessment: Checking Correctness Of Derivations And Theory:**

I carefully checked the derivations and theory.

**Review Assessment: Checking Correctness Of Experiments:**

I carefully checked the experiments.

**Review Assessment: Thoroughness In Paper Reading:**

I read the paper at least twice and used my best judgement in assessing the paper.

---

> ### Author Response · Authors · 2019-11-15
> **Response to Reviewer #1 (Part 1)**
>
> Thank you for your thoughtful and constructive review. We answer your questions below.
>
> Q1:  I'm not convinced by their motivation -- the divergence itself is less significant. If you really care about the flexibility, the Wasserstein distance may be a better choice.
>
> A1: We want to emphasize that the divergence metric essentially defines a variational inference (VI) algorithm, therefore divergence metric is a crucial factor of making VI successful. Studying the divergence in VI is an important problem in the literature such as [1,2,3,4,5], and they all have shown that the divergence plays an important role in improving VI’s performance. Our work is situated with this line of work that improves the variational inference by improving the divergence measure.
>
> This paper addresses divergence learning via meta-learning. Our divergence measure is learnable and can be selected in an automatic fashion for a certain type of tasks, which enables the advances of using a flexible family. For the choice of divergence family, we agree that the Wasserstein distance with learnable transportation cost is also very flexible. However, parameterizing transportation cost in $\theta$ space is much more involved (e.g. in [6] the cost function is parameterized by an auto-encoder.), especially when $\theta$ is of high dimensions (e.g. in BNNs). Instead for $f$-divergences $f(t)$ is an $\mathbb{R} \rightarrow \mathbb{R}$ function regardless of $\theta$ dimension, therefore it is much easier to learn with a small neural network. As shown in the paper, we are able to efficiently learn the convex $f$ function in $f$-divergence.
>
> We agree with the reviewer that the model and variational distribution choices are also important. However, we argue that different likelihood design is a modeling choice, not an inference choice. Also, the selection of variational distribution is orthogonal to the choice of the inference algorithm, and for VI the algorithm is determined by the divergence design. Our work focuses on learning the inference algorithm, and the proposed method applies to any model design and variational distribution choice.
>
> Q2: Choosing the alpha or f is a hyper-parameter search problem to me. It is impractical to collect M tasks and search for hyper-parameters according to some meta-losses.
>
> A2: The setting in the paper follows a typical meta-learning setting [7, 8]: we have several training tasks and we train a learner on them to gain common knowledge which will generalize well on the future tasks. The learner in our method is either the divergence (meta-D) or the divergence and the model initialization (meta-D&phi). This meta-learning setting is practical as demonstrated in many previous work [7, 15, 16], including meta-learning for Bayesian inference [8]. Also, this is needed in many real-world applications such as a recommender platform that needs to provide recommender service to many different companies.
>
> Besides, learning f-divergence cannot be solved by hyperparameter search because it is parameterized by a deep neural network. It will be very inefficient to optimize this neural network by hyperparameter search. Our methods instead optimize it using an optimizer such as SGD or ADAM which is clearly more efficient.
>
> Q3: The experimental part is weak. It is not the typical way that people choose to evaluate VI methods. I suggest to evaluate on standard VAE/BNN benchmarks.
>
> A3: The experiments in this paper follow the meta-learning setup as stated in Section 3, which is consistent with the experimental setup in meta-learning research [7,8,15,16] with the aim to reflect the real-world application needs. With such standard meta-learning tasks construction such as the BNN one, we can compare our results with related work where only meta-learning of the model parameters are used (e.g. MAML [7]). Apart from the recommender system with partial VAE in the submitted version of the paper, we additionally provide the experimental results on another real-world dataset, MNIST. Please see the response “Changes in the Revised Paper” for details. These experimental settings highly align with real-world applications where many datasets/tasks from different companies are available for a shared machine learning platform.

---

> > ### Author Response · Authors · 2019-11-15
> > **Response to Reviewer #1 (Part 2)**
> >
> > Major comments:
> > Q4: $p(\theta, D)$ is not possible to compute for BNNs and VAEs. More details on how to perform VI with alpha-divergence and f-divergence should be covered.
> >
> > A4: For $\alpha$-divergence, $p(\theta, D)$ is computed by $p(D | \theta) p(\theta)$ which is tractable for BNN and VAEs, as shown in [1].
> >
> > For f-divergence, the density ratio in f-divergence becomes $\frac{p(\theta | D)}{q_{\phi}(\theta)} = \frac{p(D | \theta)p(\theta)}{q_{\phi}(\theta)p(D)}$. We estimate $p(D)$ through importance sampling and MC approximation: $p(D) = E_{\theta\sim p(\theta)} [p(D | \theta)] = E_{\theta\sim q_{\phi}(\theta)} [\frac{p(D | \theta)p(\theta)}{q_{\phi}(\theta)}] \approx \frac{1}{K}\sum_k^K \frac{p(D | \theta_k)p(\theta_k)}{q_{\phi}(\theta_k)}$ where $\theta_k \sim q_{\phi}(\theta)$. After doing this, the density ratio becomes $\frac{p(\theta_k | D)}{q_{\phi}(\theta_k)} = \frac{p(D | \theta_k)p(\theta_k)}{q_{\phi}(\theta_k)}\bigg/ \frac{1}{K}\sum_k^K \frac{p(D | \theta_k)p(\theta_k)}{q_{\phi}(\theta_k)}$ which can be regarded as a self-normalized estimator, similar to the normalization importance weight in [1]. A self-normalized estimator generally helps stabilize the training especially at the beginning. We added more details about performing VI with $\alpha$ and f-divergence in Section 3 and Appendix.A in the paper.
> >
> > Q5: Can you let the meta-loss to be equal to $D_{\eta}$? I couldn't see the point why you want to learn a f-divergence while setting the meta-loss to be $D_{0.5}$.
> >
> > A5: The meta-loss cannot be equal to $D_{\eta}$. $D_{\eta}$ is the current divergence and is the thing we want to optimize. The meta-loss reflects our preference and gives guidance about how to update the parameters $\eta$. Therefore the algorithm will not be able to learn by setting the meta-loss to be $D_{\eta}$. The standard meta-learning method cannot be directly applied to learn divergence. To solve this problem, the key insight we have is that the updated variational parameters are essentially dependant on the inner loop loss function. We encode our preference into another loss function called the meta-loss to evaluate the updated variational parameters. Thus when we take the gradient of the meta-loss, the gradient is able to flow back to the parameters of inner loss function through the updated variational parameters. By doing this, we enable the learning of the inner loss function which is the divergence in our case. Our method is related to the work of meta-learning a loss function [13, 14]. This line of work aims at learning a loss function and thus will use different loss functions for the inner and outer loop.
> >
> > The goal of setting the meta-loss to be $D_{0.5}$ is to verify that our method is able to learn a good divergence. When the meta-loss is within the divergence family, ideally the algorithm should prefer this divergence. This setting allows us to directly evaluate the learned divergence by comparing it with the known preferred divergence. We have shown in the paper that the learned divergence from our methods with $\alpha$-divergence or f-divergence, is indeed very close to the known preferred divergence which demonstrates the effectiveness of our methods.
> >
> > Q6: The details of Bayesian optimization (BO) baseline should be elaborated. In fact, it is possible to choose the best alpha for each task using BO. Should this be compared?
> >
> > A6: We used the same data of the training tasks for BO. Specifically, in MoG experiments, the objective function that BO wants to minimize is the meta-loss ($D_{0.5}$ or TV). Every time BO selects an $\alpha$, we train 10 models with that $alpha$-divergence on the support sets of 10 training tasks respectively and get the mean of the meta-loss on the query sets of the 10 training tasks. The acquisition function is upper confidence bound with kappa 0.1. Each time the model is trained for 2000 iterations.
> >
> > It is possible to choose the best $\alpha$ for each test task by BO, i.e. every time we have a new test task we run BO to select the $\alpha$. However, by doing this, we are not able to extract any common knowledge from the previous tasks and running BO for each task could be very expensive. We did not include this baseline because it does not satisfy the meta-learning setting and the cost will be much higher than our methods. We think the setup of BO in the paper is a more reasonable and fair baseline for our methods.

---

> > > ### Author Response · Authors · 2019-11-15
> > > **Response to Reviewer #1 (Part 3)**
> > >
> > > Q7: What does VB mean in Section 4.2? If VB uses KL rather than alpha-divergence or f-divergence, the outperformance may only suggest that KL is insufficient there.
> > >
> > > A7: Yes, VB means VI with KL divergence. It is true that the outperformance may suggest that KL is insufficient. In fact, KL may not be the best for most applications in practice. The difficulty of using other divergence is how to select a suitable divergence. In practice, a rich family often degenerates to a simple divergence family due to the difficulty of selection (e.g. the difficulty of defining the convex function in f-divergence). The experimental results indicate that our work is able to learn a better divergence that solves the insufficiency of KL.
> > >
> > > Q8: Have you considered input convex neural networks (Amos et al. 2016) for implementing f?
> > >
> > > A8: The input convex neural networks (ICNN) (Amos et al. 2016) is an interesting alternative to parameterize the convex function in f-divergence. We use the proposed method to parameterize f-divergence due to its simplicity from an elegant math derivation. And it does not have any requirements for the architecture of neural networks in contrast with ICNN. We would be happy to study the effect of this parameterization for our methods in the future.
> > >
> > > Minor comments:
> > > Q9: The proofs of Prop 1 & 2 are missing.
> > >
> > > A9: Thanks for pointing it out. We added the reference for the proofs in the paper.
> > >
> > > Q10: However, using this KL divergence for VI has been criticized for under-estimating the uncertainty..." Any reference to this?
> > >
> > > A10: We added the related references in the paper: [3,  9, 10]
> > >
> > > Q11: Figure 1: alpha is used before it is defined.
> > >
> > > A11: We revised the paper accordingly.
> > >
> > > Q12: "Other existing definitions of alpha-divergences have dis-continuous gradients at the alpha values corresponding to KL divergences" -- missing reference.
> > >
> > > A12: We added the related references in the paper: [1, 11]
> > >
> > > Q13: "When $D_{0.5}$ is in use as the meta-loss, the corresponding $\log f*''$ is analytical" -- could you elaborate on this?
> > >
> > > A13: When $D_{0.5}$ is in use as the meta-loss, ideally the learned f-divergence should be close to $D_{0.5}$. When the f-divergence is $D_{0.5}$, the $f$ function is $f(t)=\frac{t^{0.5}}{-0.5^2}$, and the analytical expression of $\log f''(t)$ is $-1.5 \log t + C$ with $C$ reflecting the scaling constant in $f$. In Figure 2, we compare the learned $\log f''(t)$ and the ground truth $-1.5 \log t + C$. We found that the learned $\log f''(t)$ is very close to $-1.5 \log t + 0.8$. This means that our method has learned the optimal divergence $D_{0.5}$ (because the definition of $f$-divergence is invariant to constant scaling of the function $f$, i.e. $f$ and $e^{0.8} \times f$ define the same divergence). We added this explanation in the paper.
> > >
> > >
> > > [1] Li et.al., Renyi divergence variational inference
> > > [2] Hernandez-Lobato et.al., Black-box α-divergence minimization
> > > [3] Wang et.al., Variational inference with tail-adaptive f-divergence
> > > [4] Bamler et.al., Perturbative black box variational inference
> > > [5] Tao et.al., Variational Inference and Model Selection with Generalized Evidence Bounds
> > > [6] Ambrogioni et.al., Wasserstein Variational Inference
> > > [7] Finn et.al., Model-Agnostic Meta-Learning for Fast Adaptation of Deep Networks
> > > [8] Gong et.al., Meta-Learning for Stochastic Gradient MCMC
> > > [9] Blei et.al., Variational Inference: A Review for Statisticians
> > > [10] Bishop, Pattern recognition and machine learning
> > > [11] Minka, Divergence measures and message passing
> > > [12] Ma et.al., Partial vae for hybrid recommender system.
> > > [13] Houthooft et.al., Evolved policy gradients
> > > [14] Xu et.al., Meta-gradient reinforcement learning
> > > [15] Finn et.al., Probabilistic Model-Agnostic Meta-Learning
> > > [16] Kim et.al., Bayesian Model-Agnostic Meta-Learning

---

### Author Response · Authors · 2019-11-15
**Changes in the Revised Paper**

We thank all reviewers for their constructive review and have revised the paper accordingly. The changes are the following.

Appendix C. Experiment on MNIST
We added an additional experiment on the real-word dataset, MNIST, to further demonstrate the effectiveness of our methods. Specifically, we trained a variational auto-encoder (VAE) on MNIST and replaced the KL divergence in VAE by our learned $\alpha$-divergence.  Each digit is regarded as a task and we let the first 5 digits (0-4) to be the training tasks and the last 5 digits (5-9) to be the test tasks. The results align with other experiments that the learned $\alpha$-divergence is better than KL, which is the standard choice.

Section 3. Meta-VI
We strengthened our motivation and added more explanation about the practicality of the setting.

Section 4. Experiments
We added more explanation to the setup of using $D_{0.5}$ as the meta-loss.

Appendix.A Computing Equation (8) in Practice
We added more details about how to compute Equation (8).

Appendix. B.2 Approximate Mixture of Gaussians
We added more details of the BO baseline.
We explained how to get the analytical expression of $\log f*''$ in Figure 2.

Reference.
We added the missing references that R1 pointed out.

---

### Decision · Program_Chairs · 2019-12-19

**Decision:**

Reject

**Comment:**

The paper proposes a meta-learning algorithm to learn the divergence measure of variational inference as well as the initialization of the variational parameters (which reduces optimization steps of VI). Improved performance by the learned divergence against hand-designed ones are empirically shown on: Gaussian mixture approximation, Bayesian neural regression, and p-VAE based recommender.
Reviewers initally raised some concerns on hyperparameters selection, weakness of experiments, and motivation for the proposed scheme. The authors responded by adding additional experiments (MNIST) as well as some new sections in their appendix about details of their method or the baselines.
The reviewers greatly appreciated the response and commonly believed that the revised version is significantly improved over the initial draft  and the improvements of the draft. As a result of that, some reviewers increased their scores. However, some of their concerns did not resolve. In particular, R1 questions the impact of the work and importance of learning divergence measure (referring to GAN or VQ-VAE for obtaining realistic samples). Also R1 finds evaluation based on MNIST unsatisfactory, as it is commonly considered as a toy dataset. To motivate the method, it is suggested that the authors think about real applications which can highlight the benefits of their method in practice. Similar concerns are shared by R2 after authors' response. In particular, R2 is not convinced about motivation and the necessity of using meta-learning for learning the divergence. I suggest authors improve on issues around motivation and support the impact of their scheme in a more practical setting.